# Potential Benefits of the Mediterranean Diet and Physical Activity in Patients with Hidradenitis Suppurativa: A Cross-Sectional Study in a Spanish Population

**DOI:** 10.3390/nu14030551

**Published:** 2022-01-27

**Authors:** Irene Lorite-Fuentes, Trinidad Montero-Vilchez, Salvador Arias-Santiago, Alejandro Molina-Leyva

**Affiliations:** 1Dermatology Department, Hospital Universitario Virgen de las Nieves, Avenida de Madrid 15, 18014 Granada, Spain; iloritef@correo.ugr.es; 2Hidradenitis Suppurativa Clinic, Dermatology, Hospital Universitario Virgen de las Nieves, 18014 Granada, Spain; tmonterov@correo.ugr.es (T.M.-V.); alejandromolinaleyva@gmail.com (A.M.-L.); 3Instituto de Investigación Biosanitaria Granada, 18014 Granada, Spain; 4European Hidradenitis Suppurativa Foundation (EHSF), 06844 Dessau-Roßlau, Germany

**Keywords:** acne inversa, hidradenitis suppurativa, inflammation, Mediterranean diet, nutrition, physical activity

## Abstract

There is scarce scientific information regarding the potential benefits of healthy lifestyles in patients with hidradenitis suppurativa (HS). The objective of this study is to explore the potential association between the adherence to a Mediterranean diet (MD), physical activity and HS severity. A cross-sectional study that included patients with HS was conducted. Disease severity was evaluated by the International Hidradenitis Suppurativa Severity Score System (IHS4) and self-reported disease activity using a Numeric Rating Scale (NRS, 0–10). The adherence to a MD was assessed by the PREvención con DIeta MEDiterránea (PREDIMED) questionnaire and the level of physical activity by the International Physical Activity questionnaire. A total of 221 patients with HS were included in our study. The adherence to a MD was average for a Spanish population. A higher adherence to a MD was associated with lower disease activity, lower self-reported Hurley and lower IHS4. The use of extra virgin olive oil as the main culinary lipid was the dietary habit that implied a lower degree of disease activity (*p* < 0.05). Regarding physical activity, both the self-reported severity and IHS4 presented an inverse association with the intensity of physical activity. The adherence to a MD and the intensity of physical activity were positively associated. The Mediterranean dietary pattern may have an impact on HS. Greater adherence to a MD is related to lower HS severity and more physical activity also correlates to lower disease severity. The MD could be an appropriate dietary pattern for patients with HS due to its anti-inflammatory properties, and combining this with increased levels of physical activity could have additional benefits.

## 1. Introduction

Hidradenitis suppurativa (HS) is a chronic, recurrent, debilitating inflammatory skin disease of the hair follicle that usually presents with lesions in the apocrine gland-bearing areas of the body, most frequently in the axillary, inguinal, submammary and anogenital regions [1]. It is characterized by the presence of nodules, abscesses, fistulae and permanent scarring [2]. These lesions cause pain, suppuration, malodors and pruritus, greatly impairing the patients’ quality of life (QoL) [3,4].

The pathophysiology underlying this complex condition has not been clearly defined. An upregulation of various cytokines, such as tumor necrosis factor alpha (TNF-α), interleukin (IL)-1, IL-17, IL-23 and other molecules, seems to be related to this inflammatory condition [5]. The primary triggering event in HS seems to be follicular occlusion, linked to aberrant keratinization and immune dysregulation [6]. Multiple genetic and environmental factors contribute to HS development, including obesity, tobacco use, immune system disorders, hormonal imbalances and microbial dysbiosis [2]. Being overweight is the most common comorbidity associated with HS and this is related to disease severity [7]. Excess body weight induces pro-inflammatory cytokine secretion and leads to mechanical stress, friction and sweating at lesion sites, promoting follicular hyperkeratosis and physical occlusion [7]. Furthermore, some studies have shown that weight loss interventions may improve HS flares [8,9].

Despite the role of obesity in HS, nutritional status is an even more essential factor for HS propagation and exacerbation [8]. Sweets, bread/pasta/rice, dairy and high-fat foods can induce disease flares, while vegetables, fruit, chicken and fish are considered to be alleviating factors [10]. Nutritional supplements, including zinc gluconate, vitamin D and riboflavin, may also decrease HS flares [11]. Moreover, it has been proposed that limiting high glycemic foods and avoiding foods containing brewer’s yeast could improve HS severity in some patients [12].

The Mediterranean diet (MD) is an anti-inflammatory dietary regime characterized by a high proportion of fruits, vegetables, legumes, cereals, bread, fish, fruit, nuts and extra virgin olive oil, and a low consumption of meat, dairy products, eggs and alcohol [13]. A good adherence to an MD reduces the risk of long-term systemic inflammation, metabolic syndrome and cardiovascular events [14,15]. Considering skin diseases, it has been observed that a good adherence to an MD can slow the progression of psoriasis, and a lower adherence to an MD is related to a more severe disease [14]. Moreover, following an MD during pregnancy is beneficial for decreasing atopy risk in newborns [16]. To date, only one study has evaluated the adherence to an MD in patients with HS [17]. It included only 41 patients and showed that a low adherence to an MD was related to a higher score on the HS Sartorius scale [17]. Nevertheless, it is still unclear how dietary modifications could impact HS evolution [18].

Physical activity may also help patients with HS to lose weight and improve the disease severity [8]. High levels of physical activity are related to a low risk of disease flares in inflammatory bowel diseases [19]. Moreover, patients with psoriasis exercise less frequently than healthy individuals, and a lower activity level is related to a higher number of psoriatic lesions and self-awareness of the disease [20,21]. To date, there is no study evaluating physical activity in HS patients, although its level is likely to decrease depending on psychosocial impairments and symptom severity [3].

Although dietary modifications and physical activity programs are often of considerable interest to HS patients and can serve as a cost-effective adjunctive therapy for HS, there is still a lack of data regarding their relationship with HS [18,22]. Thus, the objective of this study is to evaluate the association between an adherence to an MD, physical activity and HS severity.

## 2. Materials and Methods

The Materials and Methods are summarized in Figure 1.

A cross-sectional study was conducted. Participants were recruited from the HS Clinic of Hospital Universitario Virgen de las Nieves, Granada, Spain, and the Spanish Hidradenitis Suppurativa patients’ association (ASENDHI). Patients who did not sign the written consent form had a condition that made it difficult to understand the questionnaires or were ≤18 years old were excluded from the study.

The main variables were the adherence to the MD and the physical activity. The adherence to the MD was assessed by the validated PREvención con DIeta MEDiterránea (PREDIMED) questionnaire. It consisted of 14 items about the frequency of consumption of different foods commonly eaten in the Mediterranean diet. Each item scored 1 or 0. The scores ≤ 5 suggest the lowest adherence; scores from 6–9, the average adherence and the scores ≥ 10, the highest adherence [23]. Information about physical activity was collected using the short form of the International Physical Activity questionnaire [24] validated in the Spanish population [25]. It consisted of a 5-item questionnaire about the frequency, duration and intensity (vigorous and moderate) of physical activity in the last 7 days, as well as the walking and sitting times on a weekday [25]. According to their level of physical activity, the participants were classified into three groups: vigorous, moderate or slow [24].

The clinical, socio-demographic and biometric variables were recorded by means of a clinical interview and physical examination (patients from the HS clinic), or by an online self-administered questionnaire developed with Google^®^ Forms suite (patients from ASHENDI). The socio-demographic characteristics included sex, age, civil status, height, weight, body mass index (BMI) and smoking habit. The clinical features included disease duration, Hurley stage, number of affected areas, nodules, abscesses and draining tunnels count, and previous and current treatments. The Hurley stage [26] and the International Hidradenitis Suppurativa Severity Score System (IHS4) were used to assess disease severity. The Hurley stage was self-reported by the patients [27,28]. The IHS4 score was calculated in patients who attended the HS clinic by the number of nodules (multiplied by 1), plus the number of abscesses (multiplied by 2), plus the number of draining tunnels (multiplied by 4), considering IHS4 < 4 mild inflammation, 5–10 moderate inflammation and >10 severe inflammation [29]. Self-reported disease activity was assessed using a Numeric Rating Scale (NRS), with 0 representing inactive disease and 10 the highest disease activity ever experienced [27,30,31]. These scales show the subjective impact of the disease on patients, with equal or greater importance than the objective scales [32,33].

Descriptive statistics were used to evaluate the characteristics of the sample. The Shapiro–Wilk test was used to check the normality of the variables. Continuous data was expressed as the mean ± standard deviation (SD) or as the median (25th–75th percentile) and qualitative variables as the absolute (relative) frequency. The χ^2^ test or Fisher’s exact test were used to compare the qualitative data, as appropriate. The Wilcoxon test or the Student’s *t*-test for independent or paired samples were used to compare continuous data where necessary. Multivariate logistic regression analyses were performed to independently assess the potential effect of MD on disease severity. Epidemiological and statistical criteria were used to model the variable selection. Significance was set for all tests at two tails, *p* < 0.05. Statistical analyses were performed using JMP version 14.1.0 (SAS institute, Charlotte, NC, USA).

## 3. Results

### 3.1. Sociodemographic and Clinical Features of the Sample

A total of 221 patients were included in the study: 74.2% (174/221) were recruited from ASENDHI and 21.3% (47/221) from our HS clinic. There were no differences in the sex, age, body mass index (BMI), smoking habit, adherence to an MD, self-reported Hurley stage or disease duration between the two groups. A higher proportion of patients from the HS clinic were under biologic treatment, 42.11% (24/57) vs. 26.22% (43/164) *p* < 0.001, and presented a lower self-reported disease severity, 4.28 (2.89) vs. 5.94 (2.64), *p* < 0.001. The two populations were considered together for analyses.

The socio-demographic characteristics of the sample are shown in Table 1. Participants were more frequently smokers and overweighted-obesity women around 40 years of age. Regarding the clinical features, most patients self-reported Hurley II and III with a disease duration longer than 15 years. More than half of the patients had received systemic treatment, and a third of them had been treated with biologic drugs.

### 3.2. Adherence to the Mediterranean Diet

Data from the Predimed survey (Table 2) showed that the mean adherence to an MD was 7.41 (2.16). The highest proportion of affirmative answers was found in the use of extra virgin olive oil as the main culinary lipid (86.87%, 192/221) and the lowest proportion of affirmative answers was observed in glasses of wine consumption at ≥7/week (7.69%, 17/221). Most patients presented an average adherence to the MD of 66.51% (147/221), compared to a low adherence of 17.19% (38/221) and high adherence of 16.29% (36/221).

The factors potentially associated with a higher adherence to an MD were explored (Table 3). The univariate analysis showed that being older, a lower symptom intensity, a lower self-reported Hurley and a higher intensity of physical activity were associated with a higher adherence to an MD. After conducting a multivariate regression model, it was observed that a lower self-reported disease activity, lower self-reported Hurley and higher level of physical activity were independently associated with a higher adherence to the MD.

In a second stage, the items of the Predimed questionnaire potentially associated with lower self-reported disease activity were explored. The use of extra virgin olive oil as the main culinary lipid and poultry rather than red meat consumption were the dietary habits that implied a lower degree of disease activity, *p* < 0.05.

The relationship between the adherence to the MD and disease severity assessed by IHS4 was also evaluated (only in patients of the HS clinic). A higher adherence to the MD was related to lower IHS4 (β = −0.10, *p* < 0.001), Figure 2. This association remained stable after conducting a multivariate regression model adjusted by age, self-reported Hurley, disease duration and physical activity level (β = −0.04, *p* = 0.03).

### 3.3. Physical Activity

Regarding the physical activity, both the self-reported disease activity (*p* = 0.14, Figure 3) and IHS4 (*p* = 0.27, Figure 4) presented a trend towards an inverse association with the intensity of the activity regardless of an adherence to an MD.

## 4. Discussion

Inflammation plays a key role in HS pathogenesis. There are several factors that impact HS onset and evolution, including obesity, and consequently diet and physical activity [2]. This study shows that the MD may modify HS severity. Virgin olive oil could be the most important food related to lower HS severity. Physical activity levels may also be related to healthier lifestyles and, as a consequence, with a higher adherence to the MD, and they also play an anti-inflammatory role.

To date, there is only one previous study assessing the relationship between the MD and HS [17]. It observed that 41 HS patients had a lower adherence to the MD than 41 healthy controls [17]. It did not provide the mean adherence to MD in its population. In our study, the mean adherence to the MD was 7.41, reflecting an average adherence in line with the data of a healthy Spanish population [34], and the patients with moderate to severe psoriasis [35].

Regarding disease severity, Barrea et al. found that adherence to the MD negatively correlated to HS severity, evaluated by Sartoriusscore [17]. In agreement with them, we observed that the adherence to the MD is negatively associated with the IHS4 score and self-reported disease activity [29,36]. Adherence to the MD has been also associated with disease severity in psoriasis [20,34]. MD could lower systemic inflammation by different pathways. On the one hand, it has been postulated that diet can alter the gut microbiome, modulating the inflammatory response and potentially changing HS activity [10,37]. On the other hand, another possible explanation for the MD being related to lower disease severity is the reduction of chronic, systemic inflammation due to the anti-inflammatory and antioxidant properties of dietary fibers and polyphenols, and the low amount of low-refined carbohydrates. Carbohydrates increase insulin and insulin-like growth factor 1 (IGF-1) levels, which activate FOXO1 and the mTORC1. This induces cellular proliferation in the follicular epithelium and sebaceous glands and predisposes them to follicular occlusion [10,38,39]. The anti-inflammatory power of the MD can be also due to lower amounts of simple carbohydrates, total fat, food with a higher n-6/n-3 PUFAs ratio and higher complex carbohydrates, MUFA, n-3 PUFA and fiber [17]. Moreover, it has been reported that nutrition impacts immune system regulation, so diet could also play a role in lymphocyte regulation [40].

Furthermore, this study shows that vigorous physical activity is related to a higher adherence to a MD, with a trend towards lower disease activity. Previously, it has been also observed in a healthy population that the adherence to the MD is positively related to the level of physical activity [34], as both are healthy lifestyle choices and impact on weight loss. Obesity increases insulin resistance (IR), with consequent chronic hyperglycemia [41], and increased inflammatory markers and disease flares [7]. Weight reduction could help to decrease friction in the intertriginous areas and reduce insulin resistance, improving disease severity with a synergic effect in addition to the diet [41]. In fact, it has been observed that weight reduction can improve HS, mainly by decreasing lesions in the frictional areas [42,43]. In this way, lifestyle modifications, including diet and physical activity, might change the disease evolution. Moreover, diet and physical activity may impact treatment response to TNF inhibitor therapy [44].

PREDIMED (Prevención con Dieta Mediterránea), a multicenter Spanish clinical trial, carried out on a Spanish population, observed that MD supplemented with extra virgin olive oil or mixed nuts decreased the risk of developing cardiovascular events compared to dietary fat reduction only [45]. Moreover, the clinical trial PREDIMED-Plus found that a one-year lifestyle combination of physical activity and a calorie-restricted MD improved cardiovascular risk factors, including waist circumference, triglycerides and HDL cholesterol, and reduced insulin resistance, improving glycemic control and insulin sensitivity. Smoking habits, obesity, metabolic syndrome, dyskinesia, diabetes mellitus and hypertension are cardiovascular risk factors frequently present in HS patients [1,46], which increase the prevalence of cardiovascular events in these patients [47]. Therefore, dietary and physical activity recommendations could decrease the risk of cardiovascular events in HS patients.

The results of this study should be interpreted considering some limitations: (1) its cross-sectional design, which hinders causal inferences, and (2) the inclusion of patients from two different populations, which increases sample representativeness but may incur differential selection bias. Nevertheless, a comparison of the baseline characteristics between the two populations did not show differences beyond those inherent to the treatment expected between the patients coming from a specialized unit and a patient advocacy group.

## 5. Conclusions

In conclusion, this study shows that dietary patterns may impact the inflammatory severity of HS. A greater adherence to a MD, especially the use of olive oil as the main fat for cooking/eating could be related to lower HS severity. Physical activity on its own and in combination with an MD is probably also associated with additional benefits for HS patients. Dietary and physical recommendations should be encouraged in the therapeutic strategy of patients with HS, with the MD being one of the potential advisable dietary patterns. It would be interesting to develop clinical trials using different types of healthy dietary patterns and different types and grades of physical activity to recommend the most appropriate diet and exercise for HS patients.

## Figures and Tables

**Figure 1 nutrients-14-00551-f001:**
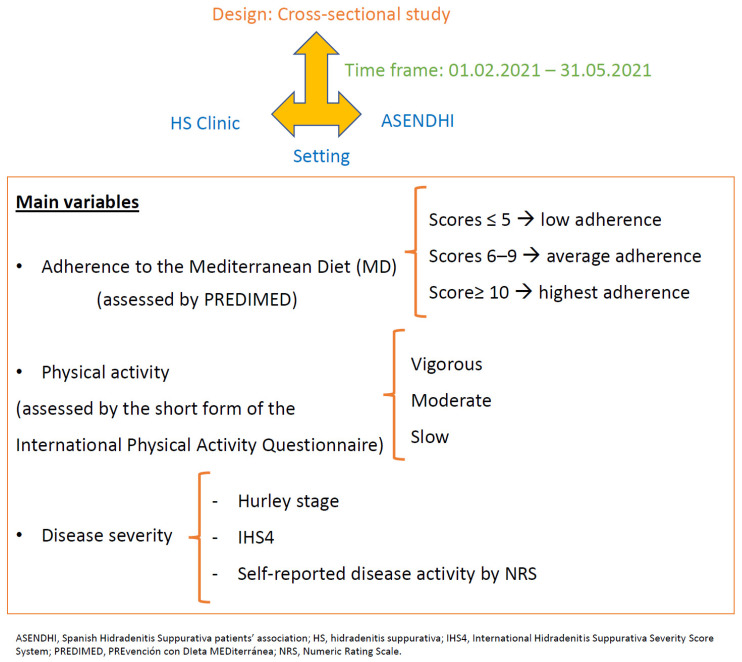
Flowchart of patients and methods.

**Figure 2 nutrients-14-00551-f002:**
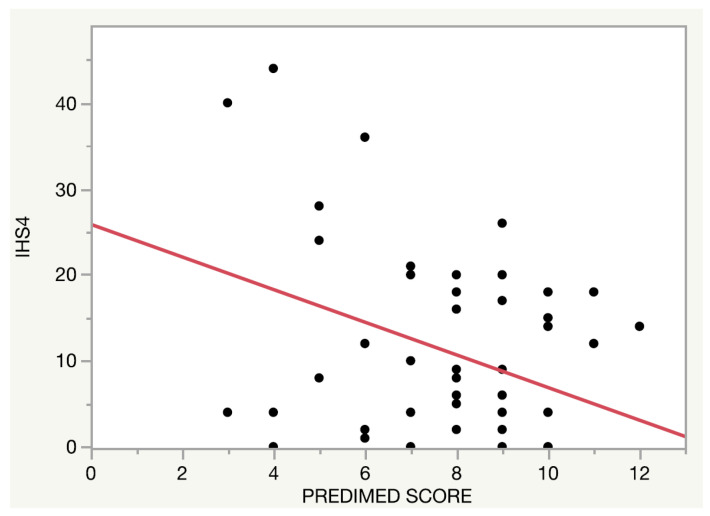
Relationship between the adherence to the Mediterranean diet and Hidradenitis Suppurativa severity.

**Figure 3 nutrients-14-00551-f003:**
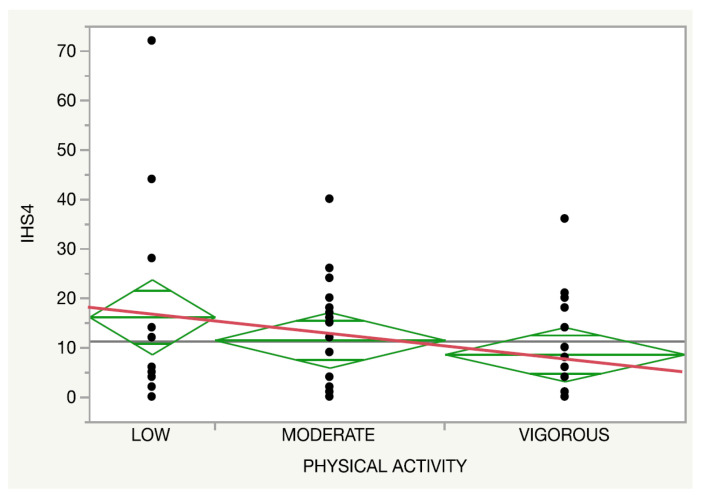
Relationship between the physical activity (red line) and the International Hidradenitis Suppurativa Severity Score System.

**Figure 4 nutrients-14-00551-f004:**
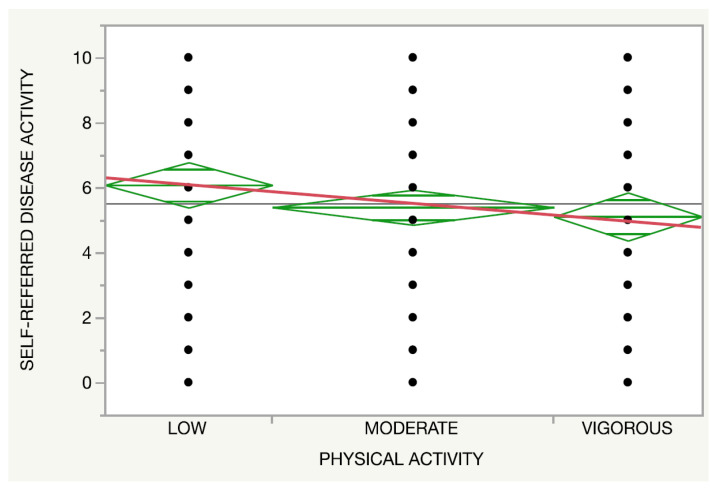
Relationship between the physical activity (red line) and self-referred disease activity.

**Table 1 nutrients-14-00551-t001:** Socio-demographic characteristics of the sample.

	*N* = 221
ASENDHI	74.20% (164/221)
Age (years)	38.36 (10.74)
Sex	
Female	73.30% (162/221)
Male	26.29% (59/221)
BMI (kg/m^2^)	29.65 (6.26)
Smoking habit (yes)	57.46% (127/221)
Number of cigarettes per day	7.60 (8.29)
Blood pressure medication (yes)	15.38% (34/221)
Cholesterol or triglyceride medication (yes)	9.95% (22/221)
Diabetes medication (yes)	8.1% (18/221)
Disease duration (years)	16.77 (9.96)
Self-reported Hurley	
I	29.41% (65/221)
II	38.91% (86/221)
III	31.67% (70/221)
Self-referred disease activity	5.51 (2.80)
Actual HS treatment	
Oral antibiotics	17.19 (38/221)
Topical antibiotics	19.00% (42/221)
Biologics	30.31% (67/221)
None	21.71% (48/221)
Systemic treatment different form antibiotics	11.68% (26/221)
IHS4 (only patients from our HS Clinic, *n* = 57)	11.31 (13.17)

ASHENDI, Asociación de Enfermos de Hidrosadenitis; BMI, body mass index; HS, Hidradenitis Suppurativa; IHS4, International Hidradenitis Suppurativa Severity Score System. Data are expressed as the relative (absolute) frequencies and means (standard deviation (SD)).

**Table 2 nutrients-14-00551-t002:** Response frequency of dietary components included in the PREDIMED questionnaire.

Predimed	%
Use of extra virgin olive oil as main culinary lipid	86.87% (192/221)
Extra virgin olive oil > 4 tablespoons	44.79% (99/221)
Vegetables ≥ 2 servings/day	52.48% (116/221)
Fruits ≥ 3 servings/day	30.76% (68/221)
Red/processed meats < 1/day	68.32% (151/221)
Butter, cream, margarine < 1/day	72.39% (160/221)
Soda drinks < 1/day	62.44% (138/221)
Wine glasses ≥ 7/week	7.69% (17/221)
Legumes ≥ 3/week	44.79% (99/221)
Fish/seafood ≥ 3/week	28.95% (64/221
Commercial sweets and confectionery ≥ 2/week	48.86% (108/221)
Tree nuts ≥ 3/week	40.27% (89/221)
Poultry rather than red meats	79.63% (176/221)
Use of sofrito sauce ≥ 2/week	73.75% (163/221)
Total Predimed punctuation	7.41 (2.16)

Percentages reflect affirmative answers’ proportion of to each question. Data are expressed as the relative (absolute) frequencies and means (standard deviation (SD).

**Table 3 nutrients-14-00551-t003:** Factors potentially associated with the adherence to a Mediterranean diet.

Variables		Univariant (Beta)	*p* *	Multivariant (Beta)	*p* **
Age		0.03 (0.01)	0.02 *	0.01 (0.01)	0.32
Sex	Male	7.40 (0.28)	0.96	-	-
	Female	7.41 (0.17)		-	-
BMI (kg/m^2^)		−0.005 (0.02)	0.82	-	-
Cigarettes/day		−0.01 (0.017)	0.44	-	-
Number of cigarettes per day		7.73 (0.37)	0.35	-	-
Blood pressure medication (yes)		7.31 (0.46)	0.82	-	-
Cholesterol or triglyceride medication (yes)		7.55 (0.51)	0.77	-	-
Diabetes medication (yes)		0.02 (0.01)	0.07	-	-
Actual treatment	Oral antibiotics	7.23 (0.35)	0.66	-	-
	Topical antibiotics	7.04 (0.33)		-	-
	Biologics	7.53 (0.26)		-	-
	None	7.54 (0.31)		-	-
	Systemic treatment different form antibiotics	7.73 (0.42)		-	-
Age of onset		0.20 (0.26)	0.43	-	-
Disease duration		0.54 (0.30)	0.07	0.01 (0.01)	0.54
Self-referred disease activity		−0.17 (0.08)	0.04	−0.11 (0.05)	0.02 **
Self-reported Hurley	I	7.86 (2.01)	0.02 *	−0.36 (0.14)	0.01 **
	II–III	7.23 (2.20)			
Physical activity level	Low–Moderate	7.09 (0.27)	0.06	0.31 (0.14)	0.02 **
	Vigorous	7.77 (0.21)			

BMI, body mass index; HS, Hidradenitis Suppurativa. Data are expressed as β coefficient (standard deviation (SD). * *p*-value after using a simple linear regression model to compare the Predimed score and the other variables. ** *p*-value after using a multivariate regression model adjusted by age, disease duration, self-referred disease activity, self-reported Hurley and physical activity. Self-reported Hurley in gathered Hurley II–III vs. I, and the physical activity level in low-moderate vs. high. Significance was set for all tests at two tails, *p* < 0.05 is considered statistically significative.

## Data Availability

Data is be available upon from to the corresponding author.

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
