# Peer review of "Potential Benefits of the Mediterranean Diet and Physical Activity in Patients with Hidradenitis Suppurativa: A Cross-Sectional Study in a Spanish Population"

_nutrients, 2022, doi:10.3390/nu14030551_

Round 1

Reviewer 1 Report

An interesting original article correlating adherence to mediterrenean diet and to physical activity with lower HS severity. Although the findings of the study don't differ from what already reported in medical literature, given the high number of patients enrolled (more than 200, ) for such a rare disease, I feel that this article should be granted publication after minor revisions:

line 40 you should add " The pathophysiology underlying this complex condition has not been clearly defined. An upregulation of various cytokines, such as tumor necrosis factor alpha (TNF-α), interleukin (IL)-1, IL-17, IL-23, and other molecules seems to be related to this inflammatory condition." and cite : doi: 10.3390/ijms21228436.

Conclusions should be expanded, better highlighting what future perspectives will open following this study results.

Thank You

Author Response

An interesting original article correlating adherence to mediterrenean diet and to physical activity with lower HS severity. Although the findings of the study don't differ from what already reported in medical literature, given the high number of patients enrolled (more than 200, ) for such a rare disease, I feel that this article should be granted publication after minor revisions:

Thank you for the comments

line 40 you should add " The pathophysiology underlying this complex condition has not been clearly defined. An upregulation of various cytokines, such as tumor necrosis factor alpha (TNF-α), interleukin (IL)-1, IL-17, IL-23, and other molecules seems to be related to this inflammatory condition." and cite : doi: 10.3390/ijms21228436.

We have included this sentence and this citation.

Conclusions should be expanded, better highlighting what future perspectives will open following this study results.

We have expanded the conclusions and have added information about future perspectives. The following sentence has been added: It would be interesting to develop clinical trials using different types of healthy dietary patterns and different types and grades of physical activity to recommend the most appropriate diet and exercise for HS patients.

Thank You

Reviewer 2 Report

This is a very useful clinical research for each physician and the HS patient. Minor, minor suggestion is too re-think Materials and methds section - for the sake of space - shortening of text and combining with figure 1 could be performed. Otherwise my congratulations!  

Author Response

Thank you for the comments. We have shortened the material and methods section following your recommendation.